# Deep Voxelized Feature Maps for Self-Localization in Autonomous Driving

**DOI:** 10.3390/s23125373

**Published:** 2023-06-06

**Authors:** Yuki Endo, Shunsuke Kamijo

**Affiliations:** 1Department of Information & Communication Engineering, Graduate School of Information Science and Technology, The University of Tokyo, Tokyo 153-8505, Japan; 2The Institute of Industrial Science (IIS), The University of Tokyo, Tokyo 153-8505, Japan

**Keywords:** autonomous driving, self-localization, deep learning

## Abstract

Lane-level self-localization is essential for autonomous driving. Point cloud maps are typically used for self-localization but are known to be redundant. Deep features produced by neural networks can be used as a map, but their simple utilization could lead to corruption in large environments. This paper proposes a practical map format using deep features. We propose voxelized deep feature maps for self-localization, consisting of deep features defined in small regions. The self-localization algorithm proposed in this paper considers per-voxel residual and reassignment of scan points in each optimization iteration, which could result in accurate results. Our experiments compared point cloud maps, feature maps, and the proposed map from the self-localization accuracy and efficiency perspective. As a result, more accurate and lane-level self-localization was achieved with the proposed voxelized deep feature map, even with a smaller storage requirement compared with the other map formats.

## 1. Introduction

Accurate self-localization in any environment is a crucial problem in autonomous driving. High-definition (HD) maps and light detection and ranging (LiDAR) are commonly used for lane-level self-localization. This map-based self-localization method estimates the pose of the self-driving vehicle by fitting the surrounding HD map components and the current LiDAR scan.

HD maps typically contain geometrical information about the target environment in the form of a set of three-dimensional points, referred to as a point cloud map. Map-based self-localization can be formulated as the cross-source point cloud registration (PCR) problem between a point cloud map and a LiDAR scan. This problem estimates a rigid-body transformation (∈SE(3)) between the coordinate systems of the global map and the local LiDAR scan [1]. Algorithms for this problem include well-known iterative closest point (ICP)-related algorithms [2] and deep-learning-based methods. As simple extensions of the ICP algorithm to deep learning, deep descriptors for each point are extracted from local or global contexts [3,4,5,6,7].

Point cloud maps contain rich information about geometrical structures but are known for redundancy. For example, a wall in a point cloud map is represented by many points, whereas only a normal vector and area are essential. Many point cloud compression techniques have been proposed to preserve keypoints [8], but point cloud compression is independent of self-localization, leading to worse self-localization results. The compression of point clouds carries the risk of losing corresponding points from scans, which can significantly degrade the performance of PCR methods.

Another map format, feature maps, is known for its efficiency in achieving lane-level self-localization using lighter maps than point clouds. One such format is the normal distribution (ND) map used in normal distribution transformation (NDT) [9]. An ND map consists of normal distributions defined in voxels. However, ND-based embedding is still self-localization-independent, and self-localization can fail in specific circumstances (e.g., at intersections) [10]. The best map format suitable for self-localization is still under discussion. The concept of self-localization using a point cloud map and a feature map is summarized in Figure 1.

Thanks to today’s breakthroughs in deep learning, deep neural networks can efficiently embed point cloud information into a fixed-length feature vector [11]. PointNetLK [12] proposes a PCR algorithm with a pair of embedded feature vectors [12]. Some improvements for the optimization stability were proposed in variant works [13,14]. Notably, their embeddings were trained to minimize errors in self-localization with some datasets. Thus, the feature vectors should be suitable for self-localization and useful for map representations. However, most of the deep PCR methods have not been evaluated in a cross-source setting but only in a same-source setting [15]. To our best knowledge, no research has treated such deep feature vectors as maps in lane-level self-localization and compared point cloud maps, feature maps, and deep feature maps regarding storage size and accuracy for self-localization in real environments. Therefore, this paper explores the possibility of map-based self-localization using deep feature vectors as a map.

However, problems arise when simply using deep feature vectors as a map due to the placement of the vectors in the environment. To access map resources from arbitrary positions of the self-vehicle, the deep feature vectors representing surrounding structures must be placed at regular intervals. Coarse placement enables a larger feature vector size in a limited storage situation, but it results in areas where the map cannot provide enough surrounding structures consistent with possible scans, as shown in Figure 2a. In these areas, range differences between a map embedding and a scan can lead to inadequate self-localization results. A denser arrangement solves the above sparsity problem, but a smaller feature vector cannot sufficiently represent the entire environment.

The problem described above arises when attempting self-localization using a single feature vector. To address this, we can explore self-localization with a map consisting of multiple feature vectors. The proposed map comprises feature vectors defined within voxels that divide the environment, as depicted in Figure 2b. Since each feature vector in a voxel represents a local point cloud, embedding becomes relatively easier than with a single feature vector representing the entire environment. This approach allows for smaller feature vectors and denser arrangements in limited storage situations. However, as conventional PCR methods using deep feature vectors assume a single feature vector per point cloud, a new PCR algorithm for multiple feature vectors needs to be developed, similar to NDT [9].

Additionally, we propose an attention mechanism to dynamically weigh the voxelized feature vectors without additional data, as depicted in Figure 2c. Previous studies have shown significant differences in the importance of voxels for self-localization [10,16]. By incorporating the attention mechanism, we can dynamically estimate the importance of each voxel, enabling flexible and robust self-localization.

Experiments compare point cloud maps, single feature vector-based maps, and proposed maps from aspects of map sizes and self-localization accuracy. The KITTI VO dataset [17] and a Shinjuku Urban dataset are used for the experiments. The latter dataset is designed by us, consisting of dense point cloud maps constructed with expensive LiDARs and scans of a cheap LiDAR sensor taken on different dates for a highly urban canyon in Shinjuku, Tokyo, Japan. Notably, such a self-localization comparison of conventional and deep methods from a map format aspect has never been well studied.

In conclusion, the possibility of self-localization with maps in real environments can be discussed by using them.

Our contributions can be summarized as follows:We conduct experiments that demonstrate that existing point-cloud-dependent deep PCR methods are outperformed by feature-based methods in terms of capacity and performance for self-localization estimation.We discuss the necessity of voxelization when constructing deep-features-based maps for self-localization and propose a deep voxelized map.We propose a self-localization algorithm that utilizes the proposed map with reassignment and attention mechanisms. We demonstrate that our method outperforms previous methods in real urban environments.

## 2. Related Work

### 2.1. High-Definition Maps for Self-Localization

Conventional map formats for self-localization are mostly based on point clouds [18,19]. Many methods for performing self-localization using point cloud maps have been proposed. They generally rely on point-to-point correspondence estimation, which is why a full point cloud map is required for self-localization.

Another format is feature maps, which consist of feature vectors extracted from point clouds. Each feature vector embeds points into high-dimensional spaces, making the number of map elements shorter than a point cloud. Feature maps are efficient for limited storage on vehicles. Such feature-map-based PCR methods include deep ones that produce intermediate high-dimensional vectors. The deep features used in their neural networks can be utilized for self-localization as a map. As discussed in the following section, some deep PCR methods utilize deep features directly for accurate registration. This paper discusses a map with deep features for self-localization.

### 2.2. Self-Localization with Point Cloud Maps

Iterative closest point (ICP) [2] is one of the pioneering works in PCR and map-based self-localization using point clouds. Many variations of ICP have been proposed to improve local minimum problems and noise robustness [20,21].

On the other hand, deep-learning-based PCR methods have been proposed using point clouds. Common sense in such methods is to estimate point-to-point correspondences via deep descriptor matching [4]. Some variations were proposed to improve the matching performance, such as coarse-to-fine matching [5] and graph-structure-based matching [6,7]. In DeepGMR [22], the EM algorithm is used to cluster the input point clouds, which reduces the difficulty of point-to-point matching by formulating the problem as a matching problem between clusters instead of corresponding points. The common assumption is that all methods assume a pair of point clouds as input. Once correspondences are estimated, the best rigid-body transformation between the pair of point clouds is analytically solved using singular value decomposition (SVD) [2].

Deep PCR methods with point clouds have emerged successfully in recent years and can be applied to self-localization [18]. However, correspondence-based methods assume corresponding points between two point clouds, which can be easily corrupted, especially in the cross-source setting. In fact, it has been reported that the accuracy of such methods decreases in the cross-source setting [15].

### 2.3. Self-Localization with Feature Maps

Besides point cloud maps, more sophisticated map formats, such as feature maps, have been used in self-localization.

One of the feature maps is a normal distribution (ND) map, which defines normal distributions inside voxels that divide environments at regular intervals. Normal distribution transformation (NDT) is the most commonly used method for utilizing normal distribution maps for self-localization. NDT was originally proposed by Biber et al. in 2D space [23] and was later extended to 3D by Magnusson [9]. In addition, some improvements to the NDT algorithm have been proposed [24]. Recently, we proposed a map that places NDs only on major structures in urban areas, such as buildings and poles [25]. ND map-based self-localization is accurate and requires only little data storage compared with point clouds. However, there are analysis results suggesting that NDT can be inaccurate in certain situations due to a lack of structural features in surrounding environments [10].

### 2.4. Self-Localization with Deep Feature Maps

Although not maps for self-localization, several deep PCR methods have been proposed that use high-dimensional feature vectors as a representation of point clouds. PointNetLK introduced a rigid-body transformation optimization on a feature space with a pair of feature vectors [12,13]. FMR [14] showed experimentally that such optimization-based methods could be applied to cross-source PCR with a reconstruction loss. In contrast to point-correspondence-based methods, feature-vector-based methods cannot solve PCR analytically. The rigid-body transformation is iteratively optimized to minimize the distance between the pair of feature vectors.

In the case of self-localization, the generation of feature vectors appears to learn an efficient representation of a map’s point cloud for self-localization, suggesting that they could be used for constructing maps and self-localization. However, to the best of our knowledge, exploring the use of a deep-feature-based map format for self-localization is under discussion.

Ref. [26] proposed deep localization to construct a single global feature as a map to estimate poses using sensor data. However, representing the entire environment with a single global feature is challenging, and the authors only use it as observations for an extended Kalman filter (EKF). It has been proposed in some works to use deep-feature-based maps as descriptors to identify similar targets in a database using sensor data [27]. However, these works evaluate their results based on recall and precision for retrieval, and accurate self-localization is not considered with the features.

In our experiments, we found that simply using a single feature vector as a map does not work well for self-localization. Instead, we propose a method that utilizes multiple small feature vectors to divide the map, which improves the accuracy of self-localization.

## 3. Deep Voxelized Feature Maps and Self-Localization

This section describes the proposed voxelized feature maps and how to utilize them in self-localization. Moreover, our attention-based optimization is introduced.

### 3.1. Discussion of Deep Feature Maps

As discussed before, FMR only uses a single feature vector as a map representation. However, problems arise when considering the placement of feature vectors in an environment.

The sparse arrangement of feature vectors results in positions lacking the necessary surrounding information for scans, as shown in Figure 2a. Range gaps between maps and LiDAR scans at such positions result in inadequate pose results, since they try to minimize the distance between their features. In contrast, dense arrangements increase map size. While a dense arrangement with small feature vectors can also be considered, small feature vectors may fail to embed the whole geometric surrounding structure, which is needed for accurate self-localization. Rather than a single-feature vector-based map, a multiple-feature vector-based map is considered better for self-localization.

### 3.2. Deep Voxelized Feature Maps

Multiple feature vectors allow for a more focused region in each vector, which can capture local geometric structures more accurately. This enables a dense arrangement of small feature vectors, as shown in Figure 2b. We refer to this type of map as a “voxelized deep feature map”, which consists of voxels containing deep feature vectors. To utilize the voxelized deep feature map for self-localization, a feature-metric optimization-based algorithm that considers multiple feature vectors as a map is necessary. FMR assumes a pair of single-feature vector maps as input, which is insufficient for accurate self-localization in complex environments.

#### 3.2.1. Difference from NDT

The resulting maps are similar to the ND maps used in NDT [9], a nondeep method. However, each voxel in the proposed map is larger and contains higher-dimensional parameters compared with NDT.

We observed that a larger voxel size could achieve accurate results in rotation. However, a larger voxel size also results in worse translation accuracy in NDT because there are too few parameters to embed points inside such larger voxels.

In our proposed map format, accurate estimation of both translation and rotation is possible with the larger voxel size and high-dimensional feature vectors. More complex shapes can now be represented, allowing for larger voxel sizes. Increasing the voxel size and adjusting the size of feature vectors makes it possible to create maps with smaller capacities than ND maps.

#### 3.2.2. Difference from Voxelization in PointNetLK-Revisited

A deep PCR method using voxels has been proposed in PointNetLK-Revisited [13]. This algorithm is called PointNetLK-Rev-Vox, as it is a part of the PointNetLK-Revisited paper. However, in their optimization process, they only treated multiple feature vectors as a single vector with a summation. While this method may be useful for rough estimation, it may lose local geometry. If the input point cloud is large and has wide local appearance varieties, a more accurate optimization can be achieved with per-voxel residual computation.

Moreover, PointNetLK-Rev-Vox takes a pair of voxelized feature vectors as input. The result highly depends on the initial voxelization, and if the true transformation is large, the assigned voxels may have large gaps. This is particularly significant with large voxels, which is the case in both PointNetLK-Revisited-Vox and our proposed algorithm. In the proposed algorithm, we formulate the optimization similarly to distribution-to-point (D2P) registration, like NDT. The proposed method computes the per-point assignment of a LiDAR scan for map voxels in each iteration. A deep feature vector for scan points inside a voxel is estimated, and a residual vector is computed.

In our experiments, we observed that the results of PointNetLK-Revisited-Vox are worse than those of the proposed method.

## 4. PCR via Deep-Feature-Metric Optimization

Before introducing the proposed self-localization algorithm, this section first discusses feature-metric optimization. The basis of this optimization method is mostly derived from PointNetLK [12]. In deep-feature-metric optimization, a neural network produces deep feature vectors from input point clouds. Then, a 3D rigid-body transformation between the pair of inputs is iteratively optimized to minimize the distance in the feature vector space.

One difference from PointNetLK is that we formulate the problem as forward- compositional (FC) [28] instead of inverse-compositional (IC), because we do not want to store large Jacobian matrices in maps. Additionally, the IC assumes that the pair of point clouds looks the same at first-order approximation, which is corrupted in the cross-source setting.

### 4.1. Preliminary

#### 4.1.1. Rigid-Body Transformation

We define a self-vehicle pose as $T\in \mathrm{SE}\left(3\right)\subset R^{4\times 4}$, a 3D rigid-body transformation from a local (i.e., LiDAR) coordinate system into the world (i.e., map) coordinate system.

We parameterize a 3D rigid-body transformation T with a 6D vector $\xi ={\left(\xi_{i}\right)}_{i=1,2,\ldots ,6}\in R^{6}$ in optimization. ξ corresponds to the coefficients of a T representation on Lie Algebra. A transformation T can be drawn as the following equation:(1)$$T=:\mathrm{Exp}\left(\xi \right)=\exp\left(\sum_{i=1}^{6}\xi_{i}e_{i}^{\wedge }\right)$$
where Exp is a shortcut to directly map a 6D vector ξ to SE(3). ${\left\{e_{i}^{\wedge }\right\}}_{i=1,2,\ldots ,6}$ are the se(3) generators. Readers can refer to the SE(3) parameterization for optimization in [12,13,29].

A ∘:SE(3)×SE(3)→SE(3) shows a compositional operator on SE(3).

#### 4.1.2. Point Cloud

Let us denote a point cloud by $P:={\left\{p_{j}\right\}}_{j=1,2,\cdots ,N},p_{j}\in R^{3}$. A dot product $\cdot :\left(\mathrm{SE}\left(3\right),R^{N\times 3}\right)\to R^{N\times 3}$ shows a 3D rigid-body transformation to a point cloud P.

### 4.2. Problem Definition

The goal is to optimize a self-vehicle pose T∈SE(3) to minimize Euclidean distances between a map feature and a feature of transformed scan points. The optimization goal at a time-step *k* can be drawn with a neural network $f:R^{*\times 3}\to R^{D}$ and an initial transformation $T^{\left(k\right)}$ as the following equation:(2)$$\min_{\xi \in R^{6}}E\left(\xi \right)$$
where
(3)$$E\left(\xi \right)={∥f\left(\left(T^{\left(k\right)}\circ \mathrm{Exp}\left(\xi \right)\right)\cdot P_{s}\right)-f\left(P_{m}\right)∥}_{2}^{2}$$$P_{\left\{s,m\right\}}\in R^{\{M,N\}\times 3}$ are LiDAR scan and map point clouds to be registered, respectively.

It can be linearized with the Jacobian matrix $J:=\partial f\left(\left(T^{\left(k\right)}\circ \mathrm{Exp}\left(\xi \right)\right)\cdot P_{s}\right)/\partial \xi$ of a neural network’s output with respect to a pose perturbation as follows:(4)$$E\left(\xi \right)\approx {∥f\left(T^{\left(k\right)}\cdot P_{s}\right)+J\xi -f\left(P_{m}\right)∥}_{2}^{2}$$

The Jacobian matrix can be estimated via numerical approximation as follows:(5)$$\begin{array}{ccc} J_{j} & \approx & \frac{f\left(\left(T^{\left(k\right)}\circ \mathrm{Exp}\left(\epsilon_{j}e_{j}\right)\right)\cdot P_{s}\right)-f\left(P_{m}\right)}{\epsilon_{j}} \end{array}$$
where $J_{j}$ means the *j*th row of the Jacobian matrix J, and $\epsilon_{j}$ is a small value and set to 0.01 in our experiments, like PointNetLK.

An analytical Jacobian matrix computation has been proposed [13]. However, the computation is highly memory-consuming, so we adopt numerical approximation for Jacobian matrix computation.

### 4.3. Feature Extraction

The feature extraction process is similar to that of PointNetLK [12]. Specifically, we use the “global feature” in PointNet as the feature produced from input points. Furthermore, we omit the T-Nets in PointNet to ensure that the output feature is variant in SE(3), as is done in PointNetLK.

### 4.4. Optimization

The SE(3) optimization on the feature space is carried out via the Levenberg–Marquardt (LM) method. The LM method utilizes the estimated Jacobian matrix J to update the current pose $T^{\left(k\right)}$.

The update rule can be derived with a dumping factor λ∈R as follows:(6)$$\begin{array}{cc} \; & \delta \xi =-{\left(J^{⊤}J+\lambda I\right)}^{-1}J^{⊤}r \end{array}$$(7)$$\begin{array}{cc} \; & T^{\left(k+1\right)}=T^{\left(k\right)}\circ \mathrm{Exp}\left(\delta \xi \right) \end{array}$$
where $r\in R^{D}$ is a residual vector for the cost function, $T^{\left(k+1\right)}$ is the estimated rigid-body transformation for the next step k+1, and λ∈R is a dumping factor.

In the normal LM method, the damping factor is scheduled discretely. However, for the backpropagation of deep learning, damping factors for each step are estimated with residual values of the cost function, as in [14,30]. We adopted a simple multilayer perceptron (MLP) for the damping factor estimation.

## 5. Self-Localization

Here, we summarize the proposed algorithm utilizing a deep voxelized feature map. An overview of the whole algorithm is shown in Figure 3.

### 5.1. Per-Voxel Residual Computation

Like normal distribution transformation (NDT) [9] nondeep methods, we compute residuals between a map voxel and its corresponding LiDAR scan points. Specifically, we assign each transformed LiDAR scan point to the nearest map voxel at each iteration in the LM method. Then, we estimate a scan feature for the voxel from the assigned scan points using a neural network and compute residual vectors with map and scan features for each voxel. This process is repeated for each iteration of optimization.

To update the pose parameters, we concatenate the residual vectors and the Jacobian matrices of all voxels as follows:(8)$$\begin{array}{ccc} r & = & {\left[r_{1}^{⊤}\;\cdots \;r_{V}^{⊤}\right]}^{⊤} \end{array}$$(9)$$\begin{array}{ccc} J & = & {\left[J_{1}^{⊤}\;\cdots \;J_{V}^{⊤}\right]}^{⊤} \end{array}$$

Then, the pose parameter update in Equation (Equation 6) is computed and applied to the current estimation using the new Jacobian matrix.

### 5.2. Attention-Based Feature Aggregation

Inspired by the successful Transformer work [31], we can introduce the attention mechanism to consider how each voxel contributes to self-localization for each iteration. Keys $k_{v}\in R^{D_{a}}$ and queries $q\in R^{D_{a}}$ are estimated by two MLPs, $g_{k}$ and $g_{q}$, from the map feature $f_{m}^{\left(v\right)}$ for a voxel *v* and the corresponding scan feature $f_{s}^{\left(v\right)}$.
(10)$$\begin{array}{ccc} a_{v} & = & \mathrm{softmax}\left(\frac{k_{v}^{⊤}q}{t\sqrt{D}}\right) \end{array}$$
(11)$$\begin{array}{ccc} k_{v} & = & g_{k}\left(f_{m}^{\left(v\right)}\right) \end{array}$$
(12)$$\begin{array}{ccc} q & = & \mathrm{pool}({\left\{g_{q}\left(f_{s}^{\left(v\right)}\right)\right\}}_{v=1,2,\ldots ,V}) \end{array}$$
where t∈R is a learnable temperature parameter to suppress rapid changes in attention levels. The initial value of t=1.0 was adopted in our implementation. We adopted the average pooling as the pool operator.

The estimated attention $a_{v}$ is then applied to the residual vector $r_{v}$. The final update to the pose parameter with estimated attention is shown below.
(13)$$\delta \xi =-{\left(J^{⊤}AJ+\lambda I\right)}^{-1}J^{⊤}Ar$$
where $A:=\mathrm{diag}(\left[a_{1},\ldots ,a_{V}\right])$. This kind of weighted LM update has already been used in some research, and noise robustness was reported [32].

Note that for attentional feature aggregation, we do not need to store additional information in the map and can use existing feature vectors to compute attention.

### 5.3. Implementation

#### 5.3.1. Loss Function

The total loss function for training the neural network is defined as the following equation.
(14)$$L=∥T_{gt}^{-1}T_{est}-I_{4\times 4}{∥}_{F}$$
where $T_{est}$ is estimated transformation, and $T_{gt}$ is ground truth transformation.

#### 5.3.2. Networks

The PointNet [11] without point/feature transformation layers was used for feature extraction from point clouds. Additionally, layer normalization [33] was used instead of batch normalization in the original PointNet.

For attention, a two-layer MLP with ReLU activation was used for the feature conversion networks $g_{\left\{k,q\right\}}:R^{D}\to R^{D_{a}}$. Note that no additional storage sizes are required to compute attention, since the network uses the same map features as the optimization process.

A one-layer MLP with a ReLU activation was used for the dumping factor estimation in the LM method, similar to BA-Net [30]. The dumping factor is estimated from a residual vector.

## 6. Experiments

The experiments compare self-localization errors using both point cloud maps and feature maps. Similar to real-world scenarios, the feature vectors and point clouds are placed in the environment prior to evaluation. During inference, the surrounding map components within a certain range (e.g., 100 m) are gathered from an initial pose, and point cloud registration is performed with the corresponding LiDAR scan.

### 6.1. Targets for Comparison

We adopted ICP [2] and NDT [9] as nondeep methods. In addition, we evaluated the deep-learning-based point cloud registration methods DeepGMR [22] and DCP [6], which utilize point cloud maps and are correspondence-based methods. We also evaluated FMR [14], a single deep feature vector-based method for feature maps. As a baseline of our method, we evaluated PointNetLK-Rev-Vox [13].

### 6.2. Datasets

We used two datasets for the experiments: the KITTI Visual Odometry (VO) and an original dataset named the Shinjuku Urban dataset. For dataset construction, we first make a point cloud map from LiDAR scans for mapping with ground-truth poses. LiDARs for mapping are typically expensive and observe much denser point clouds compared with ones for self-localization. After mapping, self-localization is carried out with a LiDAR for inference (typically cheaper than for mapping).

Table 1 shows how many pairs are included in training and testing. Five initial poses were sampled for a pair of the surrounding map and scan. Thus, the number of samples is multiplied five times for each pair. LiDAR scans at 5.0 m intervals are used from the original sequences to test differently shaped scans.

#### 6.2.1. KITTI Visual Odometry Dataset

KITTI VO is a well-known dataset for simultaneous localization and mapping (SLAM). This dataset includes sequential LiDAR scans and their ground-truth poses.

We construct a point cloud map using the sequential scan data with ground-truth poses. Self-localization is carried out with a LiDAR scan in the sequence and the accumulated scan map. With such construction, a LiDAR scan is almost an exact subset of the point cloud. Point cloud registration is easy, because there are almost exact pairs of point-to-point correspondence.

#### 6.2.2. Shinjuku Urban Dataset

A real-world dataset called Shinjuku Urban Dataset was used to register in real environments. We constructed the dataset [10,16]. The target area is a highly urban environment in Japan, which is typically challenging for self-localization because of many obstacles and the complexity of the environment [16].

This dataset comprises dense point cloud maps and LiDAR scans for an urban environment in Shinjuku, Tokyo, Japan. The dense point cloud maps are constructed with expensive LiDAR sensors. In contrast, the scan data for self-localization was taken independently with a cheap LiDAR three years after mapping. Due to the gaps in time between the mapping and the self-localization scan, such situations will often happen, making the Shinjuku dataset more challenging for self-localization than the constructed map-to-scan dataset from the KITTI VO dataset. Readers can refer to the previous papers to know how to construct the maps and ground truth in detail [10,16].

The dataset consisted of 5 km paths, of which 2.5 km were used for training and validation, and the remaining 2.5 km were used for testing. A bird-eye view is shown in Figure 4.

### 6.3. Implementations

Almost all settings, including the optimizer and learning rate schedule, were based on PointNetLK [12] and FMR [14]. In comparison, all the deep learning models were trained for 100 epochs, and the batch sizes for training were adjusted not to exceed 24 GB of VRAM usage.

For ICP and NDT, we utilized the implementations provided by PCL [34]. For the deep learning methods, we utilized the authors’ implementations. We reimplemented voxelization and replaced inverse composition (IC) with forward composition (FC) for PointNetLK-Rev-Vox [13] to match the proposed method. The settings of PointNetLK-Rev-Vox, except residual computation and attention mechanism, were set the same way as in the proposed method.

In general, settings in self-localization, global navigation satellite systems (GNSS), and inertial navigation systems (INS) are utilized for initial guess estimation. As a possible range of initial guesses, we sampled 5 initial poses for each target scan from a uniform distribution between 0.0 m and 0.8 m for translation in meters and a uniform distribution between 0.0 deg and 30.0 deg for the yaw angle. This means that we generated five trials of self-localization for a LiDAR scan to ensure generalizability to different initial poses. We then gathered surrounding map components (i.e., points or voxels) within a specific range and used them for self-localization. This means that points within the range were used in ICP, DCP, and DeepGMR; the nearest neighbor feature vector was used in FMR, and voxels within the range were used in NDT and the proposed method. We used a range of 40.0 m for the KITTI VO dataset and 100.0 m for the Shinjuku Urban dataset, based on the LiDAR ranges.

The three feature vector placement intervals of 5.0 m, 10.0 m, and 50.0 m were simulated for the single-feature map in FMR. The larger interval allows for a larger feature vector dimension. Assuming a storage size of 20.0 kB for feature vectors, the dimension of the feature vectors was adjusted accordingly. For the proposed method, the dimension was fixed at 128. Note that the proposed map size for each dataset is different, because the number of voxels in each dataset can vary depending on the removal of voxels with fewer points from the map.

### 6.4. Error Comparison with Full Data

First, we compare the errors using full map data. With full point cloud maps, the accuracy of point cloud map-based methods will be better than with compressed ones, and it is considered to be the maximum accuracy. We carry out a performance comparison based on the root mean square errors (RMSEs) in rotation (deg) and translation (m).

#### 6.4.1. Results

The results are shown in Table 2 and Table 3. The number of input points for all methods is set to 10,000, except for DCP [6], where the point cloud was subsampled to half the number of points (i.e., 5000) due to out-of-memory (OOM) issues with 10,000 points.

For DCP, DeepGMR, and FMR, which are deep PCR methods, the translation error is relatively high compared with the other methods. DeepGMR achieved better rotation results. In the case of NDT, the larger the voxel size, the lower the rotation errors, but the higher the translation errors. The proposed method achieves much better results in both translation and rotation, even with a much smaller map size.

The proposed method’s attention mechanism improves translation and rotation accuracy.

#### 6.4.2. Discussion of Point-Cloud-Based Methods

The results suggest that deep point-cloud-based methods may struggle to estimate point-to-point correspondences for the map-to-scan setting accurately. This is particularly true for the Shinjuku Urban dataset, where translation errors were observed to increase. These methods estimate point-to-point correspondences once, but there is no guarantee that corresponding points exist between a scan and a map. In contrast, iterative estimation methods such as ICP and the proposed method can better handle the cross-source difficulty and achieve more accurate results.

While DeepGMR is able to estimate point cloud clusters, resulting in precise rotation results, its estimation is conducted in three-dimensional space, which can lead to lower accuracy in translation. In contrast, the proposed method’s optimization is conducted in a high-dimensional space, resulting in more accurate translation results.

#### 6.4.3. Discussion of Feature-Based Methods

The results of feature-based methods, such as NDT, FMR, and the proposed method, are discussed here. As shown in the tables, the larger the voxel size, the more accurate the rotation results. With a large voxel size, they can converge to an adequate transformation from a global context perspective. However, translation accuracy is insufficient for lane-level self-localization. It is considered that the six-dimensional parameters of the voxels in NDT are not enough for translation estimation. In contrast, the proposed method utilizes high-dimensional features defined in larger regions, resulting in accurate results for both rotation and translation.

Regarding FMR, the proposed method divides the whole area of neighbor map components into regular-sized voxels, which can ease the difficulty of point cloud embedding for training networks, resulting in more accurate results.

PointNetLK-Rev-Vox is not able to handle large transformations between a map and a scan. However, in the proposed method, assigning per-voxel scan points can reduce the impact of initial voxelization, resulting in accurate results, particularly in rotation. Moreover, the pairwise residual computation can capture differences in local neighbor voxels, leading to precise results. However, residual summation in PointNetLK-Rev-Vox may lead to a loss of local context during optimization.

### 6.5. Qualitative Evaluation

A qualitative evaluation was conducted using sequential scans for self-localization. For FMR [14], feature vectors with 256 dimensions were placed in 5.0 m intervals as a map, and self-localization was evaluated. As the target scans are sequential, the problem of range gaps between a LiDAR scan and a map, as shown in Figure 2a, is expected to appear. In contrast, our proposed method consists of voxelized features and is expected to be stable throughout the sequential scans.

The results are shown in Figure 5.

FMR shows highly accurate results regularly but collapsed results for positions between them. This concludes that the single-feature-based map collapsed in sequential scans in practice. The results of our method are relatively stable throughout the sequence, and consistent fitting parts can be found in registered point clouds.

## 7. Conclusions

There are two main streams of map formats for self-localization today: point cloud maps and feature maps. While feature maps, such as ND maps, are a more sophisticated format, their embedding is human-designed. Deep learning may assist with embedding feature maps; however, our experiments have shown that utilizing conventional deep embedding for mapping causes poor accuracy problems.

This paper proposes a deep voxelized feature map as a sophisticated map format for self-localization with deep learning. A neural network encodes a point cloud map into a set of feature vectors within voxels. During inference, neighbor voxels from the self-vehicle are gathered and utilized in iterative optimization. By incorporating a method that minimizes distance errors for each voxel, the proposed method handles local information rigorously and obtains more accurate results. Additionally, we propose a method for dynamically estimating the importance of voxels with an attention mechanism to deal with surrounding environmental information.

Experiments showed that the proposed map-based self-localization method outperformed point-cloud-based and single-feature vector-based methods in terms of accuracy and storage size. Deep point-cloud-based methods [2,6,22] were found to be particularly inadequate for estimating translations. Similarly, a single-feature vector-based method [14] could not estimate poses accurately.

We also compared our method with NDT [9], a conventional voxelization-based method. NDT achieved better performance than the point-cloud-based and single-feature-based methods, even with a smaller map size. It was observed that in NDT, larger voxel sizes resulted in more accurate rotation estimation but less accurate translation estimation. This suggests that larger voxels can be useful for considering global contexts and improving rotation estimation, but the six-dimensional parameters used in NDT are not enough for accurate translation estimation. In contrast, the proposed method achieved more accurate results than NDT in both rotation and translation by utilizing larger voxels and high-dimensional features.

The attention-based weighting of feature vectors also improves accuracy, indicating that there are important differences among voxels that can be used for more efficient data compression.

Importantly, due to the limited storage size and the lane-level requirement for translation accuracy, the proposed deep voxelized feature format is a good choice for self-localization maps. However, the proposed method needs to be trained for each specific environment and voxel size setting. To address this issue, we could learn common features in a specific environment and apply transfer learning to the network. Additionally, we could consider adaptive voxel size adjustment techniques such as those used in DeepGMR [22]. These are potential avenues for future work. The deep voxelized feature map will be crucial for map-based self-localization in future autonomous driving.

## Figures and Tables

**Figure 1 sensors-23-05373-f001:**
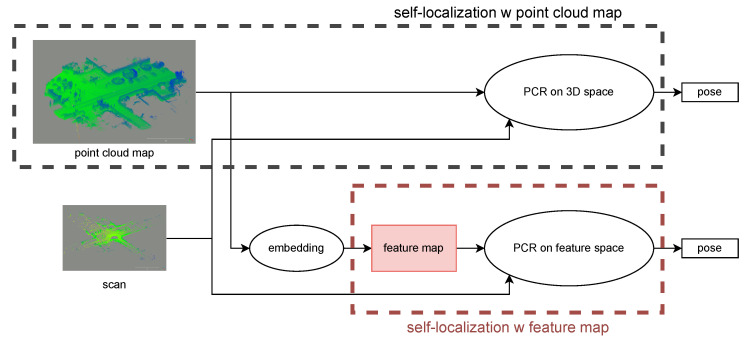
Concept of self-localization with a point cloud map and a feature map.

**Figure 2 sensors-23-05373-f002:**
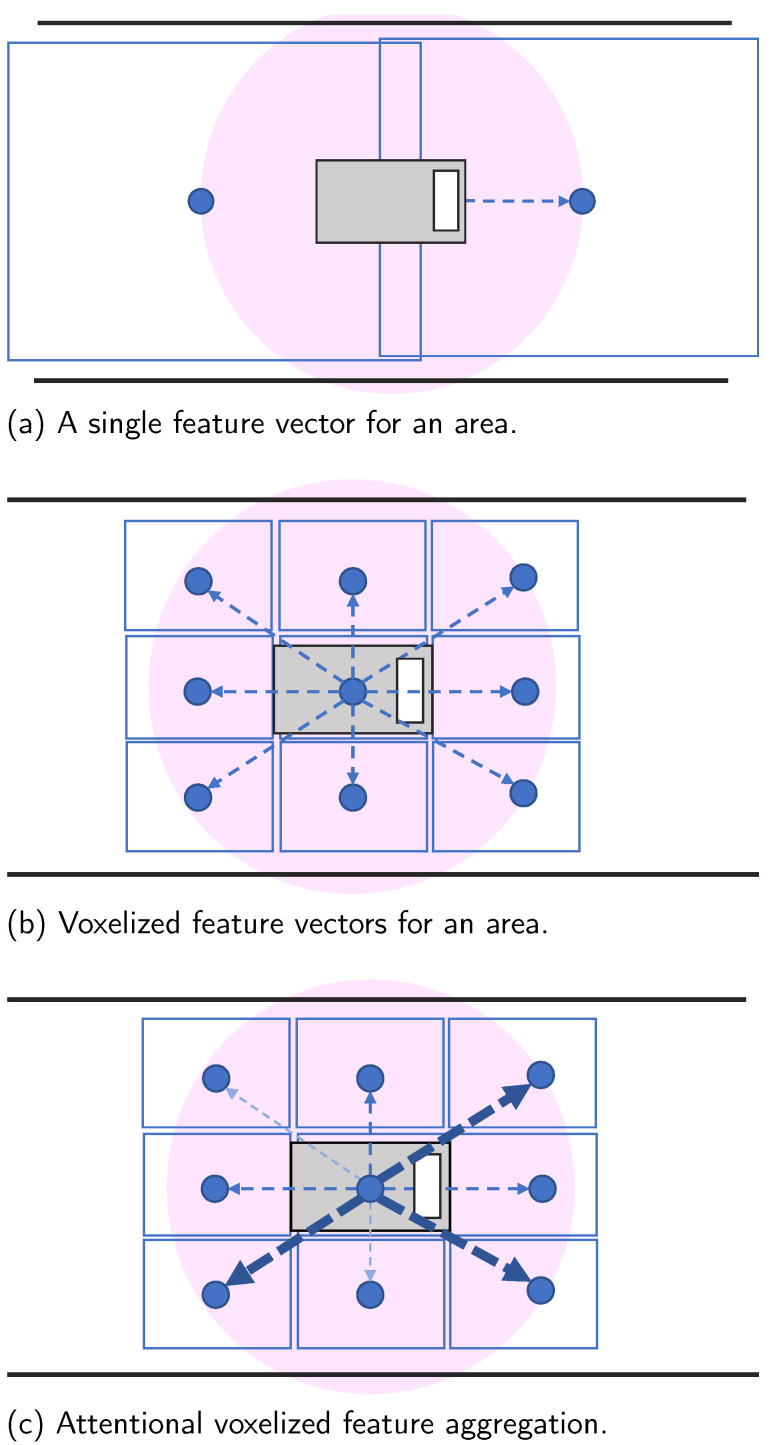
Concept of the proposed map formats: The blue rectangles represent the range covered by a feature vector, the blue dots indicate the position of a feature vector, and the blue dotted arrows indicate focusing on the surrounding features. The pink circles indicate the range of a LiDAR scan from the vehicle. Note that the position of feature vectors is fixed before driving since it is a map, but the LiDAR scan position changes with time.

**Figure 3 sensors-23-05373-f003:**
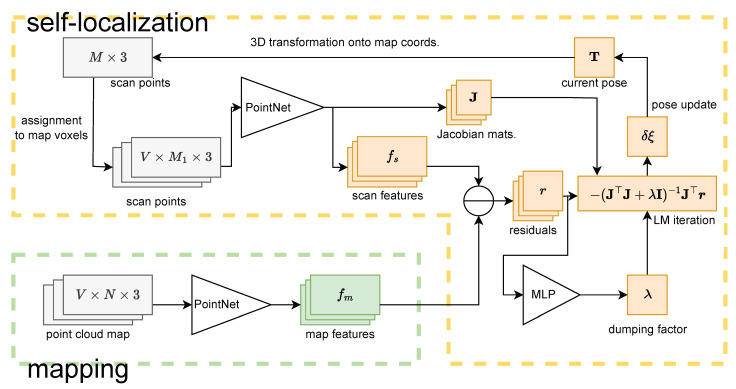
An overview of the algorithm using a deep voxelized feature map.

**Figure 4 sensors-23-05373-f004:**
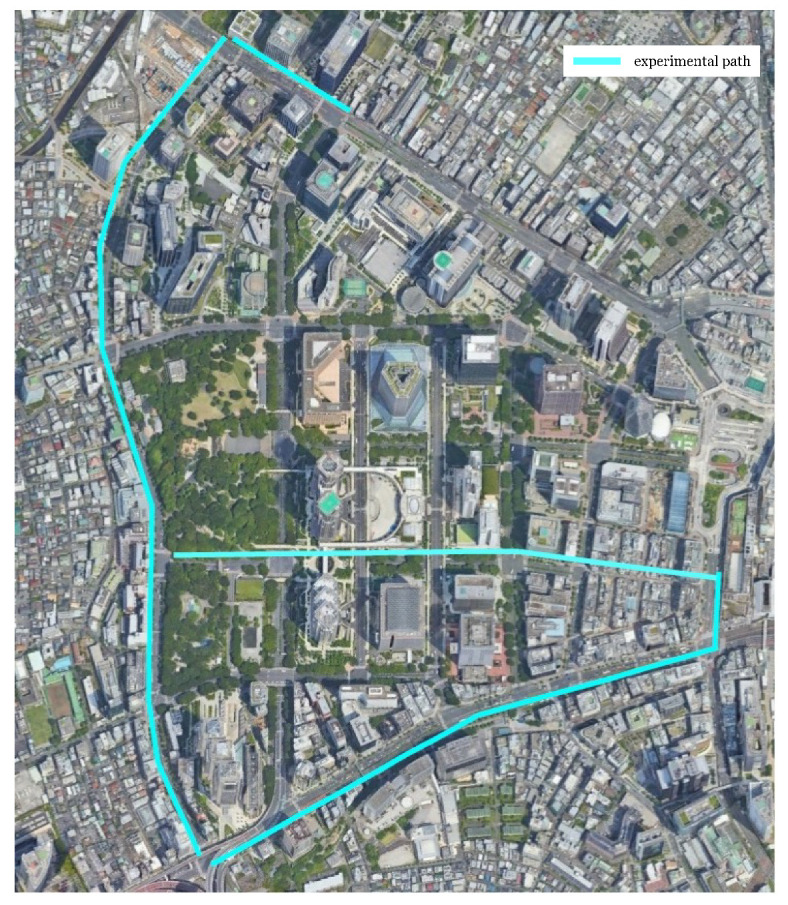
An bird-eye view of Shinjuku Urban Dataset: The light-blue lines indicate the path for the experiments.

**Figure 5 sensors-23-05373-f005:**
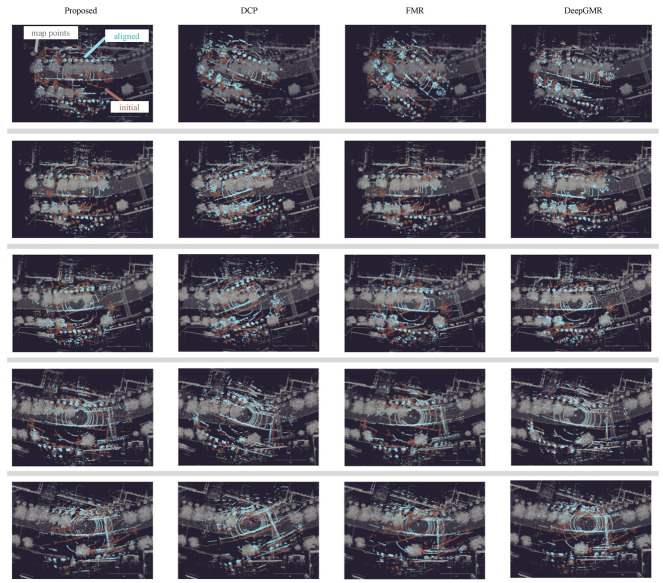
Qualitative results: A brown point cloud shows an initial LiDAR scan, a sky blue point cloud shows the registered point cloud, and a gray point cloud shows the map point cloud. A map point cloud was downsampled for visualization.

**Table 1 sensors-23-05373-t001:** Contents in Datasets.

Dataset	For Training	For Testing
KITTI VO Dataset	2535	5160
Shinjuku Urban Dataset	2320	1490

**Table 2 sensors-23-05373-t002:** Self-localization error comparison on KITTI VO Dataset.

Method	Format	Map Size (kB)/100 m	Mean		Median	
			Rot	Trans	Rot	Trans
ICP	pc	120.0 kB (10,000 pts.)	13.01	0.41	0.45	0.14
DeepGMR [22]	pc	120.0 kB (10,000 pts.)	6.23	1.26	0.98	0.43
DCP [6]	pc	60.0 kB (5000 pts.)	14.79	0.98	3.55	0.41
FMR [14]	feature	20.0 kB (D = 256/5.0 m)	11.77	1.25	2.59	0.41
		20.0 kB (D = 512/10.0 m)	12.86	1.69	1.66	0.47
		20.0 kB (D = 2560/50.0 m)	20.87	4.12	1.38	0.70
NDT(P2D) [9]	feature	37.0 kB (VS * = 2.0 m)	12.96	0.44	0.68	0.23
		17.4 kB (VS = 3.0 m)	12.17	0.50	0.45	0.15
		9.5 kB (VS = 4.0 m)	11.29	0.65	0.33	0.13
		5.5 kB (VS = 5.0 m)	10.25	0.82	0.31	0.13
PointNetLK-Rev-Vox (FC) [13]	feature	8.9 kB (VS = 20.0 m, D = 128)	14.29	0.77	2.20	0.45
ours (w/o attention)	feature	8.9 kB (VS = 20.0 m, D = 128)	8.32	0.27	1.17	0.14
ours (w attention)	feature	8.9 kB (VS = 20.0 m, D = 128)	7.98	0.27	0.99	0.14

* VS means voxel size, and D means feature dimension.

**Table 3 sensors-23-05373-t003:** Self-localization error comparison on Shinjuku Urban Dataset.

Method	Format	Map Size (kB)/100 m	Mean		Median	
			Rot	Trans	Rot	Trans
ICP	pc	120.0 kB (10,000 pts.)	13.11	0.45	0.61	0.17
DeepGMR [22]	pc	120.0 kB (10,000 pts.)	6.90	1.88	3.64	0.98
DCP [6]	pc	60.0 kB (5000 pts.)	13.51	1.74	2.63	1.33
FMR [14]	feature	20.0 kB (D = 256/5.0 m)	13.95	1.49	3.33	0.59
		20.0 kB (D = 512/10.0 m)	8.92	1.71	1.41	0.43
		20.0 kB (D = 2560/50.0 m)	13.02	5.29	1.47	0.73
NDT(P2D) [9]	feature	100.6 kB (VS * = 2.0 m)	13.03	0.48	0.76	0.24
		48.0 kB (VS = 3.0 m)	12.39	0.53	0.54	0.17
		26.7 kB (VS = 4.0 m)	11.56	0.62	0.36	0.13
		16.9 kB (VS = 5.0 m)	10.32	0.78	0.31	0.12
PointNetLK-Rev-Vox (FC) [13]	feature	17.6 kB (VS = 20.0 m, D = 128)	13.69	0.74	2.68	0.48
ours (w/o attention)	feature	17.6 kB (VS = 20.0 m, D = 128)	8.97	0.15	0.22	0.05
ours (w attention)	feature	17.6 kB (VS = 20.0 m, D = 128)	7.33	0.12	0.21	0.04

* VS means voxel size, and D means feature dimension.

## Data Availability

Not applicable

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
