# Peer review of "Deep Voxelized Feature Maps for Self-Localization in Autonomous Driving"

_sensors, 2023, doi:10.3390/s23125373_

Round 1

Reviewer 1 Report

This manuscript sensors-2327993 investigates voxelized deep feature maps for self-localization, consisting of a set of deep features defined in small regions. The self-localization algorithm proposed in this paper considers per-voxel residual and re-assignment of scan points in each optimization iteration, which could result in accurate results. Our experiments compared point cloud maps, feature maps, and the proposed map from the self-localization accuracy and efficiency viewpoint. As a result, more accurate and lane-level accuracy was achieved with the proposed voxelized deep feature map, even with a small storage requirement. It was a pleasure reviewing this work and I can recommend it for publication in Sensors after a major revision. I respectfully refer the authors to my comments below.

1.         The English needs to be revised throughout. The authors should pay attention to the spelling and grammar throughout this work. I would only respectfully recommend that the authors perform this revision or seek the help of someone who can aid the authors.

2.         (References) Please adjust the style of all the references to meet the Sensors Journal requirement.

3.         (Page 8) The original figures 3, 4 and 5 are not clear. Please redraw this figure clearly. Add some word in this figure, and indicate the imaging modules.

4.         (Section I, Introduction) The reviewer suggest to revise the original statement as “Deep PCR methods with point clouds have successfully appeared in recent years and can be applied to self-localization <*>. ("Bo Yang, Yukexin Zhang, Localization and tracking of closely-spaced human targets based on infrared sensors, Infrared Physics & Technology, 2022.”)

5.         (Section I, Introduction) The reviewer suggest to revise the original statement as “… between the pair of coordinate systems of a map (global) and a LiDAR scan (local) []. (Dongbing Guo, Baoling Qi, Chunhui Wang, Fast clustering method of LiDAR point clouds from coarse-to-fine, Infrared Physics & Technology, 2023.)

6.          (Section 2 Related work) The reviewer hopes the introduction section in this paper can introduce more studies in recent years. The reviewer suggests authors don't list a lot of related tasks directly. It is better to select some representative and related literature or models to introduce with certain logic. For example, the latter model is an improvement on one aspect of the former model.

7.         The reviewer suggests to add a new paragraph in Introduction part to summary the contribution of this manuscript.

8.         (Section I, Introduction) The reviewer suggest to revise the original statement as “… thanks to today’s deep learning breakthroughs, deep neural networks [1] ..”. (Shuai Hao, Shan Gao, et al. Anchor-free infrared pedestrian detection based on cross-scale feature fusion and hierarchical attention mechanism,  Infrared Physics & Technology, 2023.)

9.         (Page 2) The original statement “Even, machine learning communities has accepted that progress in a particular application domain is considerably accelerated when a large number of datasets are collected in unconstrained conditions. <*>, …”. (Facial expression recognition method with multi-label distribution learning for non-verbal behavior understanding in the classroom. Infrared Physics & Technology 2021, 112, 103594.;; Learning fusion feature representation for garbage image classification model in human–robot interaction. Infrared Physics & Technology 2023, 128, 104457.)

10.     (Figures 4-5) Experimental pictures or tables should be described and the results should be analyzed in the picture description so that readers can clearly know the meaning without looking at the body.

11.     (Tables) All the values in this table should be with same data accuracy. The number of data after the decimal point are the same. Please check other Tables.

12.     The authors are suggested to add some experiments with the methods proposed in other literatures, then compare these results with yours, rather than just comparing the methods proposed by yourself on different models.

13.        Discuss the pros and cons of the proposed model.

My overall impression of this manuscript is that it is in general well-organized. The work seems interesting and the technical contributions are solid. I would like to check the revised manuscript again.

Reviewer 2 Report

There are still a few issues that should be further explained, which can be found as follows:

1. In abstract, please the authors described the main result in this work, not tell reader the current these drawbacks, Maybe that can be moved in introduction.

2. Literature review is too short as the current status of the related work is missing. If possible, please discuss a map format with deep features.

3. Could you give the efficiency and complexity of the self-localization algorithm proposed in this paper

4. I did not get the meaning of eq.(1), please give more explanation. Without theoretical analysis, there is no way to prove the correctness of the simulation results.

Author Response

"Please see the attachment

Reviewer 3 Report

One of the issues it raises is the use of voxels, in figure 3 the voxelization does not appear. There is also nothing said about the way PointNet is used, it is supposed to use the "Point Features" extension for "Segmentation Network". Since the input is limited to 1024 or 1024k samples with a relatively small k, the voxelization is well suited. None of this is detailed.

The omission of the second T-net of the network is indicated, which performs a feature transform nx64. I don't know if it's possible to indicate why.

I understand that the PointNet network will be pre-trained. Against what data set?

It would be good to clarify if the general process starts from a region of the map or selects the optimal region within a neighborhood range. Or it just determines the optimal rigid body transformation for a region of the map and a scan from the LiDAR.

Round 2

Reviewer 1 Report

The revised manuscript is improved compared to the former version. My previous comments are well addressed, and the presentation is improved significantly. The composition pattern and some other ideas are well elaborated, making them clearer. Overall, I tend to accept this manuscript.